# To Inject or to Reject? The Body Image Perception among Aesthetic Dermatology Patients

**DOI:** 10.3390/jcm12010172

**Published:** 2022-12-26

**Authors:** Ida Yurtsever, Łukasz Matusiak, Jacek Cezary Szepietowski

**Affiliations:** 1Dermatology and Aesthetic Medicine, 50-266 Wroclaw, Poland; 2Department of Dermatology, Venereology and Allergology, Wroclaw Medical University, 50-368 Wroclaw, Poland

**Keywords:** body image, BDD, aesthetic medicine, cosmetic dermatology, self-esteem, body satisfaction, body appreciation

## Abstract

Background: Nowadays, aesthetic dermatology treatment has become not only physical beautification but also it can have positive effects on patients’ mental health. Body dysmorphic disorder can be the reason for treatment dissatisfaction. In the general population, the prevalence of BDD is 1.9% and it is more common among cosmetic patients. The aim of this study was to conduct the most comprehensive evaluation of body image and BDD among aesthetic patients. Methods: We recruited a group of 412 individuals, who were asked to complete 6 different on-line questionnaires concerning self-image, i.e., COPS, AAI, FAS, BAS-2, BSQ-16, and RSES. Results: The prevalence of BDD ranged from 7.28% to 11.17%, depending on the screening tool that was used. Our research revealed that BDD susceptibility, body image, body appreciation, and self-esteem were strongly interrelated (*p* < 0.001). A higher BMI was a risk factor for BDD, body dissatisfaction, and depreciation. The financial status markedly influenced all of the features. A history of psychiatric treatment influenced the risk of BDD, body satisfaction, body appreciation, and self-esteem. A history of cosmetic procedures and treatment satisfaction had no impact on the obtained results. Conclusions: Improving recognition of body dissatisfaction among aesthetic patients is very important. The psychometric assessment of patients before cosmetic treatment could be of help in choosing the appropriate approach.

## 1. Introduction

The growing popularity of aesthetic dermatology treatment and easy accessibility to the aesthetic market has led to the situation that undergoing aesthetic procedures has become not only for physical beautification but also it can have positive effects on patients’ mental health. A report that was conducted in 2018 revealed that in Poland 600,000 individuals admitted to undergoing aesthetic treatment. A total of 65% of individuals that were undergoing aesthetic procedures were residents of larger cities, 94% had a higher or secondary education, 75% were professionally active, and 59% of them had good monthly income (above GDP per capita). The main motives to undergo aesthetic treatment that were reported by the responders were the desire to improve well-being and increase attractiveness, self-confidence, and self-acceptance. While most of the women were driven to act by dissatisfaction with their appearance, a third of the men went to an aesthetic medicine office at the instigation of a loved one. A total of 96% of aesthetic dermatology patients are happy with the procedures that are performed [1]. However, the amount of unhappy individuals after aesthetic treatment is constantly increasing. Did the procedure go wrong, or were the primary expectations unreal? Where is the line between innocent beautification and unhealthy, numerous face/body modifications? Would evaluating body image or screening for body dysmorphic disorder (BDD) not be a reasonable solution before injecting? These are the questions that are worth asking and we will try to answer them in our work.

Body image is defined as a multidimensional construct, consisting of self-perception and self-attitude (concerning physical appearance). Body image has two aspects: body dissatisfaction, which is conceptualized as a poor or negative body image versus positive body image, a.k.a. body appreciation [2]. A poor body image includes preoccupation with one’s body, body shame, and body dissatisfaction. It may influence self-esteem and quality of life leading to depression, eating and sexual disorders, and emotional instability and poor self-esteem [3,4]. Positive body image can be understood as respecting, accepting, and holding favorable opinions toward the body [2].

BDD is a mental health disorder where assumed defect in physical outlook, disables everyday life functioning [5,6,7,8,9]. It is preoccupation with thinking and behaviors connected to concerns with appearance. Enrico Morselli, the Italian psychiatrist firstly described BDD in 1891 [5]. He used the term “dysmorphophobia” which referred to the Greek word “dysmorphia”, meaning hideousness. BDD is correlated with severe suffering, constant intrusive thoughts, shame, depression, poor quality of life, social distancing, and suicide [5,6,7,8]. In general population, the prevalence of BDD is assessed as 1.9%. It is more common among cosmetic dermatology (around 9%) and cosmetic surgery patients (around 13%) [5,6,10].

It is worth highlighting that approximately 76% of BDD patients undergo aesthetic treatments in order to ‘repair’ perceived defects in their physical appearance [6]. Moreover, most of the patients with BDD presenting to aesthetic professionals do not recognize themselves as suffering from any mental health disorder [5]. Unfortunately, BDD among aesthetic patients is usually or misdiagnosed or underdiagnosed, which leads to an increased risk of treatment dissatisfaction, not to mention the delay in necessary psychiatric treatment, depression, prolonged suffering, and even suicide [7,11].

The aim of this study was to conduct the most comprehensive evaluation of body image and BDD among patients undergoing aesthetic dermatology procedures. Although some of the aspects that were analyzed in the paper have been previously studied, e.g., BDD in aesthetic patients [12,13,14] or self-esteem and body satisfaction [15], to the best of our knowledge our study encompasses the largest population of aesthetic patients ever studied to date, using multiple screening tools; there has never been such diversity of questionnaires used in one paper. We also wanted to underline the importance of the psychometric assessment, facilitating proper diagnosis, before conducing the beauty treatment.

We assumed that individuals undergoing aesthetic treatment are more susceptible to self-image/ body-image disturbances and BDD, which could led to unsatisfying results, therefore we wanted to underline the importance of the psychometric assessment, facilitating proper diagnosis, before conducing the beauty treatment. For that purpose we tried to evaluate BDD versus non BDD individuals, the body image, the body appreciation and self-esteem level in light of other aspects such as age, sex, BMI, history of psychiatric treatment, past procedures, education and financial status.

## 2. Materials and Methods

We recruited group of 412 subjects, i.e., 397 women and 15 men, aged 18–71 years (mean, 35.78 ± 7.57 years), who checked into an aesthetic clinic in order to have an aesthetic procedure in the close future (biostimulators, botulinum toxin, hyaluronic acid fillers, mesotherapy, skin resurfacing, vascular laser treatment, etc.). It is worth underlying, that the sample size of 385 subjects is the minimum number to meet the desired statistical constraints (to have a confidence level of 95% that the real value is within ±5%) and to be considered as representative for the Polish population. The detailed characteristics of the studied group is given in the Table 1. The participants were asked to complete 6 different on-line questionnaires that were all validated for Polish language, concerning body image in general. The main criterium we took into consideration, next to general availability of the questionnaires, to assess all the analyzed parameters, i.e., BDD, body image, body appreciation and self-esteem, was the availability of the validated tools in Polish language. Besides the questionnaires, the studied individuals were also asked for their demographics, self-reported financial status, education level, a history of psychiatric treatment and previous aesthetic treatment. The patients were given an iPad with Google Forms, that was previously created specifically for that purpose. Informed consent was obtained from each individual.

This project was conducted in accordance with the principles of Good Clinical Practice and the principles of the Helsinki Declaration of the World Medical Association and was approved by the Bioethical Committee of the Medical University of Wroclaw (KB number 325/2020).

The questionnaires that were applied in our study were:

The COPS (the Cosmetic Procedure Screening Questionnaire), which evaluates the unattractive characteristics with regard to BDD diagnostic criteria. The questionnaire contains 9 items which are rated from 0 points (least impaired) to 8 points (most impaired), ranging from 0 to 72 points. The score is a sum of inquiries 2 to 10. Questions 2, 3, and 5 are reversed. The higher result indicates for greater impairment. Individuals who have 40 points or more are likely to have BDD [11,16,17].

The AAI (the Appearance Anxiety Inventory) was developed primarily to measure the outcomes of the therapy in people with BDD. The scale encompasses 10 questions. Each item is evaluated on a 5-point Likert scale ranging from 0 points (not at all) to 4 points (all the time). The total score is obtained by summing the items. The maximum score is 40 points, and higher scores indicate greater frequency of a process. Due to the fact that the author of the AAI did not establish a clinical cut-off, we assumed the cut-off for the BDD high-risk group as 20, we took the average of two scores that were reported by Veale et al. in 2014, where a clinical sample of adults with BDD had a median score of 27 points, and adults that were concerned with their appearance, 13 points [18,19].

The FAS (the Functionality Appreciation Scale) was developed to reflect appreciation of body functionality: respecting, recognizing, and honoring the capability of the body. It consists of 7 questions ranging from 0 points (no appreciation) to 5 points (maximal appreciation). The score is the average of all the items, with higher scores indicating greater appreciation of body functionality (ranges 0–5 points) [20,21].

The BAS-2 (the Body Appreciation Scale) is comprised of 10-items, answered on a 6-point scale (0–5) each. The score is the mean of the items. The higher scores indicate higher body appreciation [2,22].

The BSQ-16 (the Body Shape Questionnaire) evaluates body shape concerns that are typical of anorexia nervosa and bulimia nervosa. The questionnaire contains of 16 items, scored from 0 points to 6 points (least and most impaired, respectively), with the sum of the questions ranging from 0 points to 96 points. A cut-off of 38 points suggests mild concern with shape. The questionnaire was designed for women but can be used for men with slight wording changes following the author’s guidelines [21,23,24].

The RSES (the Rosenberg Self Esteem Scale) is one of the most used self-esteem measurements. It encompasses 10 items, evaluated on a 4-point Likert scale each (strongly agree, agree, disagree, and strongly disagree). Questions 1, 2, 4, 6, and 7 are reversed and have a positive impact. The total score ranges from 10 points to 40 points where a score less than 15 points signifies low self-esteem [25].

### Statistical Analysis

The statistical analysis was performed with Statistica 13.3 software (TIBCO Software Inc., Palo Alto, CA, USA). To examine the differences between categorical variables, the Chi-squared test was used. Differences between groups were established using the Kruskal–Wallis test and the Mann–Whitney U-test as the analyzed variables were of abnormal distribution. The Spearman’s rank correlation analysis was used to assess the strength of the relationship between two different variables. *p*-values that were less than 0.05 indicated statistical significance.

## 3. Results

Of the 415 patients that were approached, 3 (0.72%) did not complete the questionnaires, leaving the total number of 412 subjects. The group consisted of 397 women, aged 18–71 years (mean, 35.82 ± 7.61 years) and 15 men, aged 23–46 years (mean, 34.73 ± 6.62 years). All of the participants were identified as white, employed, and the majority had high self-reported socioeconomical status (83.01%, *n* = 341; i.e., good 57.52%, *n* = 237 or very good 25.49%, *n* = 105). The lowest education level was high school diploma (10.44%, *n* = 43) and higher education was stated by 89.56% of participants (*n* = 369). A history of psychiatric treatment was given as follows: antidepressants 9.95% (*n* = 41) and antipsychotics 4.13% (*n* = 17). No one reported the diagnosis of BDD in the past. A history of cosmetic procedures was reported by 343 patients (83.25%) and 315 patients (81.40%) stated mood improvement after the aesthetic treatment (387 patients completed the treatment satisfaction question, as the questionnaires were handed in after the aesthetic procedure, even if it was the first time). The detailed characteristics of the studied group is given in the Table 1.

### 3.1. BDD

There were two different questionnaires, i.e., the COPS and the AAI, that were used to screen for BDD among the participants. High risk of BDD ranged from 7.28% (*n* = 30) using the COPS to 11.17% (*n* = 46) using the AAI. The mean value of the COPS total score was assessed as 19.21 ± 12.13 points (range, 0–69 points) and the AAI total score was 11.34 ± 6.57 points (range, 0–40 points). For the need of the research interpretation, the studied individuals were divided into the BDD and the non-BDD group (Table 2). High risk BDD patients were significantly younger according to the AAI (33.50 ± 8.03 years vs. 36.06 ± 7.47 years; *p* = 0.02) than those with a lower risk of BDD; such a trend was also observed for the COPS (33.97 ± 7.45 years vs. 35.92 ± 7.57 years). The BMI of the patients that were suspected of BDD was higher than of those without the BDD suspicion (24 vs. 22; *p* = 0.02 for both questionnaires). Taking socio-economic status into consideration, according to both, the COPS and the AAI, high education to secondary education ratio was significantly lower in the BDD group than in the non-BDD group (60% vs. 93.46% and 80.43% vs. 92.35%, respectively). The financial status was reported to be higher in the non-BDD group (Table 2). Of note, psychiatric treatment was reported significantly more often in the BDD group than the non-BDD group; 41 patients reported taking antidepressant agents (COPS: 33.33% vs. 8.14%, *p* < 0.001; AAI: 21.74% vs 8.49%, *p* < 0.01), whereas 17 patients reported antipsychotic agents (COPS: 20.00% vs. 2.8%, *p* < 0.001; AAI: 15.56% vs 2.74%, *p* < 0.01). Surprisingly, a history of cosmetic procedures and treatment satisfaction was given by similar percentage of patients in both the BDD and the non-BDD groups (both questionnaires). There were no significant differences between the genders (both scales).

### 3.2. Body Appreciation

Body appreciation was estimated using the FAS and the BAS-2, which revealed comparable results. The mean value of the FAS total score was assessed as 4.1 ± 0.72 points (range 1.0–5.0 points) and the BAS-2 total score as 3.58 ± 0.84 points (range 1.2–5.0 points). Although the age differences were not correlated to the body appreciation, a higher BMI was reported as a risk factor for the lower body appreciation (r_S_ = −0.2, *p* < 0.01). Patients with a higher body appreciation stated better financial status (statistical significance was observed just for the BAS-2, *p* < 0.01; the FAS revealed a trend, *p* = 0.1). A history of psychiatric treatment was linked to lower body appreciation (antidepressant treatment vs. no treatment: FAS total score of 3.71 ± 0.96 points vs. 4.19 ± 0.74 points, *p* = 0.001, BAS-2 total score of 3.31 ± 1.08 points vs. 4.64 ± 0.88 points, respectively [*p* = 0.04]; antipsychotic treatment vs. no treatment: FAS total score of 3.47 ± 0.94 points vs. 4.18 ± 0.75 points, *p* = 0.001, BAS-2 total score of 3.0 ± 1.06 points vs. 3.63 ± 0.9 points, respectively {*p* < 0.01}). Interestingly, a history of cosmetic procedures and treatment satisfaction did not influence the body appreciation (Appendix A). Of note, a lower body appreciation was observed significantly more often in patients who were suspected of BDD than in those without BDD suspicion (*p* < 0.001) (Table 3.). No significant differences between the genders were observed.

### 3.3. Body Image

The BSQ-16 was used to evaluate body image. The mean value of the BSQ-16 total score was assessed as 42.73 ± 16.07 points (range, 2–87 points). Mild concern with shape was reported by 120 individuals (29.13%), while moderate concern was reported by 72 individuals (17.48%) and marked concern was reported by 39 individuals (9.47%). The participants with higher BMI achieved higher total scores of BSQ-16 (r_S_ = 0.3, *p* < 0.001). A history of psychiatric treatment negatively influenced body image as assessed with BSQ-16 total score (antidepressants vs. no antidepressants: 50.24 ± 20.03 points vs. 41.82 ± 15.31 points, respectively [*p* = 0.02]; antipsychotics vs. no antipsychotics: 57.53 ± 21.60 points vs. 42.04 ± 15.48 points, respectively [*p* < 0.01]). Moreover, worse financial status was reported significantly more frequently by patients with a higher risk of shape concerns (*p* < 0.01). Again, a history of cosmetic procedures and treatment satisfaction did not influence body image (Appendix A). It should be highlighted that 51.28% (20/39) of individuals with marked concern with shape were suspected of BDD significantly more often according to the COPS (*p* < 0.001) and 74.36% (29/39) of individuals according to the AAI (*p* < 0.001), (Figure 1 and Figure 2). There were no significant differences between the genders that were observed.

### 3.4. Self-Esteem

Self-esteem was assessed by the RSES. The mean value of the RSES total score was 20.98 ± 5.44 points (range, 2–30 points). A low self-esteem was reported by 48 individuals (11.65%). The lower self-esteem that was assessed with the RSES total score was dependent on a history of psychiatric treatment (antidepressants vs. no antidepressants: 18.32 ± 6.62 points vs. 21.30 ± 5.21 points, respectively [*p* < 0.01]; antipsychotics vs. no antipsychotics: 17.35 ± 5.61 points vs. 21.18 ± 5.36 points, respectively [*p* < 0.01]). The higher the financial status, the higher the self-esteem that was noted (*p* < 0.001). Once again, a history of cosmetic procedures and treatment satisfaction did not influence the self-esteem (detailed data not shown). It is worth underlining that according to the COPS, 14 out of 48 individuals (29.17%) with low self-esteem were suspected significantly more frequently for BDD (*p* < 0.001) and according to the AAI, it was 18 out of 48 individuals (37.5%) (*p* < 0.001) (Figure 3 and Figure 4). No significant differences were found between the genders.

### 3.5. Correlations

The Spearman’s rank correlation test revealed that all the studied features were significantly interrelated as follows: the patients with a higher risk of BDD had lower body image, body appreciation, and lower self-esteem than those with lower risk; lower body appreciation was observed in subjects with lower body image, lower self-esteem, and BBD suspicion. A higher self-esteem was related to higher body image, higher body appreciation, and with a lower risk of potential BBD. Poor body image was related to lower body appreciation, lower self-esteem, and higher risk of BDD (Table 3).

## 4. Discussion

The aim of this study was to examine body image perception in cosmetic dermatology patients using multiple screening tools. We assumed, based on literature, that concerns with appearance are higher in aesthetic dermatology patients than in the average population. The prevalence of BDD in our studied group ranged from 7.28% to 11.17%, depending on the screening tool that was used. Those slight differences might be the result of an imprecisely defined cut-off for the AAI in the literature. Of note, the usage of a cut-off of 20 (also assumed by us) was previously described by Mastro et al. [26]. Moreover, Veale et al. [10] in 2016 summed up the five studies on BDD in cosmetic dermatology patients (less than 152 participants each), obtaining the weighted prevalence of 9.2%, which is similar to ours; the prevalence of BDD in the general population was substantially lower and assessed as 1.9%. Marked concern with shape, which also can be defined as poor body image, was assessed in our study as 9.47% and low self-esteem was found to be 11.65%.

Our research revealed that all of the examined features (BDD susceptibility, body image, body appreciation, and self-esteem) were strongly interrelated. Obviously, BDD and body dissatisfaction (poor body image) are known to be associated with low self-esteem, shame, constant intrusive thoughts, depression, anxiety, social distancing, and poor quality of life [4,5,6,7,26,27]. Conversely, low self-esteem is known as a risk factor for poor body image and BDD [15,26,28]. The dependance between body appreciation and body image was previously described by Alleva et al. in 2014 [20].

Although the available literature describes differences in BDD, body image, body satisfaction, and self-esteem between genders [6,29], we did not find such dependencies. It might be the result of scarce representation of the male group in our research.

We observed a relationship between higher BMI and BDD, body dissatisfaction, and a lack of body appreciation. The study that was conducted by Lu-Hsu et al. [30] in 2021 also revealed a link between BMI and body satisfaction. Interestingly, we did not find any correlation between BMI and self-esteem although previous studies underlined being overweight as a factor that was associated with low self-esteem [28]. The possible explanation of this fact is the currently growing popularity of trends that are linked with a body positive attitude.

One of the most interesting findings in our study was that all of the questionnaires revealed that finance status markedly influenced all of the examined features, i.e., susceptibility to BDD, low body appreciation, poor body image, and low self-esteem. The only conclusion which could be derived from that is rather terrifying. Is socioeconomic status really so important? In previous studies, Raymore et al. [31], Valeska et al. [32], and Macêdo-Uchôa et al. [15] analyzed the relationship between socioeconomic level and self-esteem; they found a significantly higher level of self-esteem among the individuals with a higher level of finance and education. Therefore, it can be assumed that individuals from lower socioeconomic groups are more vulnerable to lower self-image and self-esteem and are thus an important target group in spreading awareness and implement prevention.

We also revealed that a history of psychiatric treatment highly influenced the risk of BDD, body satisfaction, body appreciation, and self-esteem. Previous studies described a history of depression as a known risk factor for body image disruptions and BDD due to greater sensitivity over physical appearance [3,4,5,6,7,8]. Conversely, depression can develop due to body dissatisfaction, as a result of prolonged anxiety and shame that is associated with BDD and the resulting social dysfunction [33].

It is worth underlining that a history of cosmetic procedures and treatment satisfaction did not influence any of the examined features. At the beginning of our study, we assumed whether individuals with body dissatisfaction and BDD would be dissatisfied with the received treatment, which would lead to further mood decrease. Now, there is a reason to believe that particular patients with body dissatisfaction might actually benefit from some of the aesthetic dermatology treatment. To date, the vast majority of studies have indicated that cosmetic procedures that were performed on individuals with BDD were associated with poor outcomes [7]. However, Veale at al. [34] noticed that the outcomes of the aesthetic treatment varied according to the procedure. Those findings suggested that certain cosmetic procedures are associated with better outcomes for BDD. Similarly, Crerand et al. [35] revealed a trend towards a more positive response to surgical interventions compared to less invasive treatment in terms of preoccupation with the treated body part. Moreover, Arruda Felix et al. [36] proved that patients with mild to moderate BDD may benefit from rhinoplasty.

This study has some limitations, which in our opinion do not lessen the acquired results. Firstly, the control group was not included in the research, yet the available studies on the prevalence of BDD in the general population are numerous, as well as for the Polish population [6,8,10]. Secondly, the number of male subjects was rather scarce. Although the study reflects actual situations, it was difficult to distinguish potential differences between genders. Moreover, this was a single-center study, however, it should be underlined that the sample size was sufficient to be considered representative for the whole population (to have a confidence level of 95% where the real value is within ±5% of the measured value). Finally, the studied group was not examined according to the treatment that was received. The satisfaction of the aesthetic treatment dependence on the procedure being performed could be the subject for further studies.

## 5. Conclusions

Summarizing, it is very significant to improve recognition of body dissatisfaction by all means among aesthetic dermatology patients. The psychometric assessment of each patient before cosmetic procedures may play an important role in planning the treatment.

A multidisciplinary approach and co-operation between aesthetic physicians, psychiatrists, and psychologists could be the standard of the future in determining the line between pathologic body image disturbances and healthy where one is satisfied and proud with self-image, which is required for normal development [27]. A refusal to perform aesthetic procedures and send for psychiatric treatment seems to be the best therapeutic option for some disturbed patients.

As a closure, we would like to add some positive attitude towards individuals with broadly understood body dysmorphic disorders. Being more aesthetically sensitive than general results in a greater emotional reaction to more attractive features, and the importance of beauty plays a greater role in their identity. Therefore, some BDD patients may have greater aesthetic appreciation skills, manifested in their education in art, design, or aesthetic practice itself [6].

## Figures and Tables

**Figure 1 jcm-12-00172-f001:**
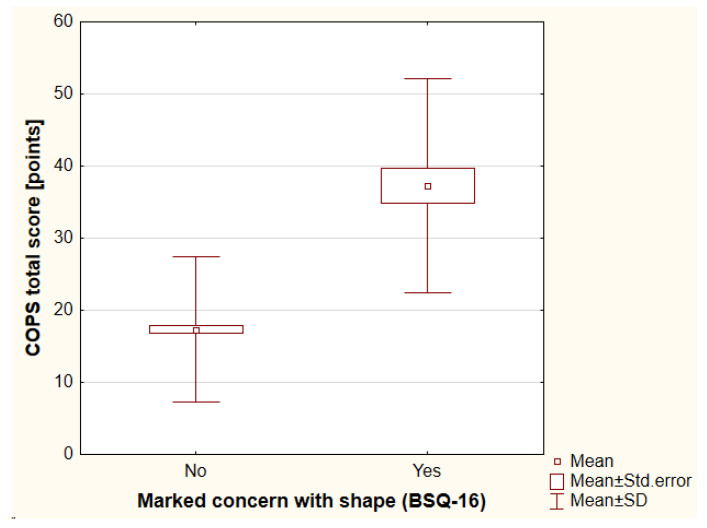
The differences in the COPS total score between individuals with marked concern with shape and without such concern. COPS—the Cosmetic Procedure Screening Questionnaire, BSQ-16—the Body Shape Questionnaire-16, SD—standard deviation.

**Figure 2 jcm-12-00172-f002:**
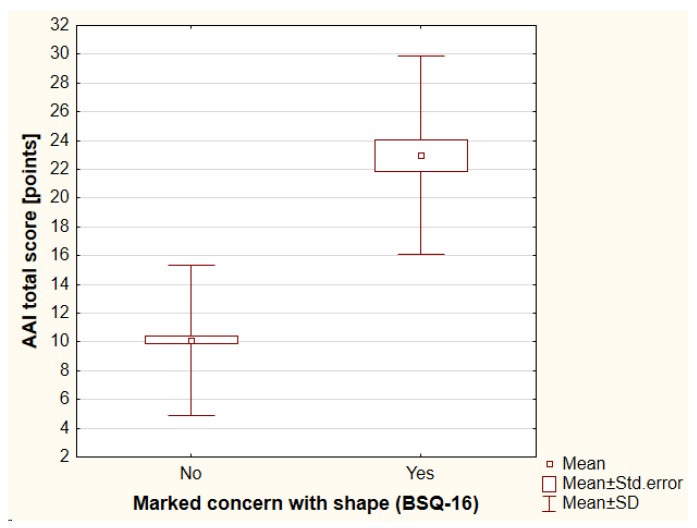
The differences in AAI total score between individuals with marked concern with shape and those without such concern. AAI—the Appearance Anxiety Inventory, BSQ-16—the Body Shape Questionnaire-16, SD—standard deviation.

**Figure 3 jcm-12-00172-f003:**
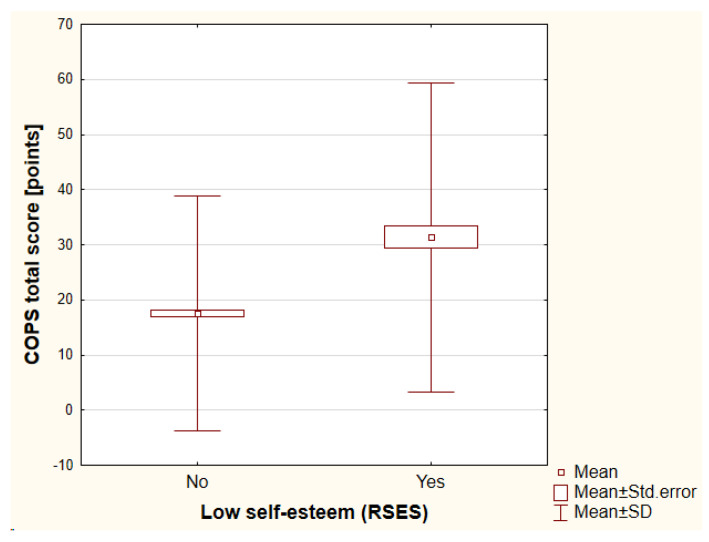
The differences in COPS total score between individuals with low self-esteem versus high self-esteem. COPS—the Cosmetic Procedure Screening Questionnaire, RSES—the Rosenberg Self Esteem Scale, SD—standard deviation.

**Figure 4 jcm-12-00172-f004:**
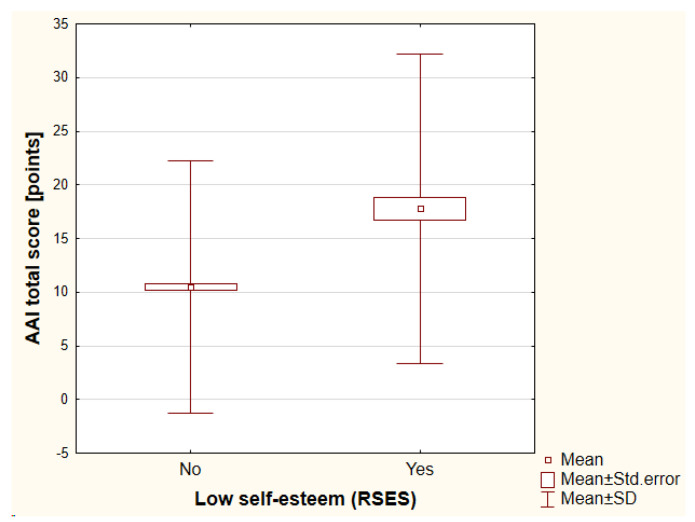
The differences in COPS total score between individuals with low self-esteem versus high self-esteem. AAI—the Appearance Anxiety Inventory, RSES—the Rosenberg Self Esteem Scale, SD—standard deviation.

**Table 1 jcm-12-00172-t001:** Characteristics of the studied group undergoing aesthetic dermatology procedures.

	All (*n* = 412)	Females (*n* = 397)	Males (*n* = 15)	*p*-Value *
**Age (mean ± SD)**	35.78 ± 7.57	35.82 ± 7.61	34.73 ± 6.62	NS
**BMI (mean ± SD)**	22.1 ± 3.95	22.1 ± 3.95	24.96 ± 2.71	<0.001
**BMI ≥ 25 (*n*)**	61 (14.81%)	54 (13.6%)	7 (46.67%)	<0.001
**BMI ≥ 30 (*n*)**	15 (3.64%)	15 (3.78%)	0 (0%)	N/A
**Antidepressants (*n*)**	41 (9.95%)	38 (9.57%)	3 (20%)	NS
**Antipsychotics (*n*)**	17 (4.13%)	15 (3.71%)	2 (13.33%)	NS
**First time AP (*n*)**	69 (16.75%)	64 (16.12%)	5 (33.33%)	NS
**Treatment satisfaction after APs (*n*)**	315 (81.40%)	306 (80.95%)	9 (75%)	NS
**APs done before (*n*)**				
needle mesotherapy	184 (44.66%)	175 (44.08%)	9 (60%)	NS
hyaluronic acid /filler/	202 (49.03%)	197 (49.62%)	5 (33.33%)	NS
botulinum toxin	188 (45.63%)	183 (46.1%)	5 (33.33%)	NS
laser	188 (45.63%)	181 (45.59%)	7 (46.67%)	NS
PDO lifting threads	16 (3.88%)	16 (4.03%)	0 (0%)	N/A
plastic surgery	48 (11.65%)	47 (11.84%)	1 (6.67%)	NS
non-invasive cosmetic procedures	214 (51.94%)	208 (52.39%)	6 (66.67%)	NS
**Education level (*n*)**				
elementary education	0 (0%)	0 (0%)	0 (0%)	<0.01
high school diploma	43 (10.44%)	38 (9.57%)	5 (33.33%)	
graduate degree	369 (89.56%)	359 (90.43%)	10 (66.67%)	
**Financial status (*n*)**				
poor	12 (2.91%)	12 (3.02%)	0 (0%)	NS
average	48 (11.65%)	48 (12.09%)	0 (0%)	
good	237 (57.52%)	228 (57.43%)	9 (60%)	
very good	115 (27.92%)	109 (27.46%)	6 (40%)	

*—differences between genders. BMI—body mass index, AP—aesthetic procedure, SD—standard deviation, NS—not significant, N/A—not applicable.

**Table 2 jcm-12-00172-t002:** Characteristics of the BDD and non-BDD groups that were undergoing aesthetic dermatology procedures, according to different screening tools.

	COPS			AAI		
	BDD (*n* = 30)	Non-BDD (*n* = 382)	*p*-Value	BDD (*n* = 46)	Non-BDD (*n* = 366)	*p*-Value
**Age (mean ± SD)**	33.97 ± 7.45	35.92 ± 7.57	NS	33.5 ± 8.03	36.06 ± 7.47	0.02
**BMI (mean ± SD)**	22.09 ± 3.89	23.9 ± 4.24	0.02	22 ± 3.21	24 ± 7.39	0.02
**Antidepressants (*n*)**	10 (33.33%)	31 (8.12%)	<0.001	10 (21.74%)	31 (8.47%)	<0.01
**Antipsychotics (*n*)**	6 (20%)	11 (2.88%)	<0.001	7 (15.22%)	10 (2.73%)	<0.001
**Non-first time AP (*n*)**	25 (83.33%)	315 (82.46%)	NS	35 (76.09%)	307 (83.88%)	NS
**Treatment satisfaction after APs (*n*)**	24 (82.76%)	291 (81.28%)	NS	36 (83.72%)	279 (81.1%)	NS
**Education level (*n*)**						
elementary education	*n*/A	N/A	<0.001	N/A	N/A	0.04
high school diploma	12 (40%)	25 (6.54%)		9 (19.57%)	28 (7.65%)	
graduate degree	18 (60%)	357 (93.46%)		37 (80.43%)	338 (92.35%)	
**Financial status (*n*)**						
poor	4 (13.33%)	8 (2.09%)	0.001	3 (6.52%)	8 (2.19%)	0.04
average	10 (33.33%)	37 (9.69%)		8 (17.39%)	40 (10.93%)	
good	8 (26.67%)	230 (60.21%)		17 (36.96%)	223 (60.92%)	
very good	8 (26.67%)	107 (28.01%)		18 (39.13%)	95 (25.95%)	

BDD—body dysmorphic disorder, COPS—the Cosmetic Procedure Screening Questionnaire, AAI the Appearance Anxiety Inventory, BMI—body mass index, AP—aesthetic procedure, SD—standard deviation, NS—not significant, N/A—not applicable.

**Table 3 jcm-12-00172-t003:** Spearman’s rank correlation coefficient (r_S_) for total scores of all questionnaires that were used in the study (*p* < 0.0001 for all correlations).

	COPS	AAI	FAS	BAS-2	BSQ-16	RSES
COPS		0.80	−0.43	−0.65	0.67	−0.56
AAI	0.8		−0.42	−0.67	0.72	−0.57
FAS	−0.43	−0.42		0.53	−0.38	0.52
BAS-2	−0.65	−0.67	0.53		−0.66	0.68
BSQ-16	0.67	0.72	−0.38	−0.66		−0.51
RSES	−0.56	−0.57	0.52	0.68	−0.51	

COPS—the Cosmetic Procedure Screening Questionnaire, AAI—the Appearance Anxiety Inventory, FAS—the Functionality Appreciation Scale, BSQ-16—the Body Shape Questionnaire-16, BAS-2—the Body Appreciation Scale, RSES—the Rosenberg Self Esteem Scale.

## Data Availability

Data supporting the reported results can be obtained on request, e-mail: ida.yurtsever@gmail.com.

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
