# Peer review of "To Inject or to Reject? The Body Image Perception among Aesthetic Dermatology Patients"

_jcm, 2022, doi:10.3390/jcm12010172_

Round 1

Reviewer 1 Report

Dear authors,

I found particularly interesting the investigation of this topic, which is still underexplored in the literature. In general, I appreciate the implications this study is seeking to inspire. 

However, I have to underline some critical points that require a revision:

Abstract

·      I would not say that aesthetic dermatology treatment is a sort of psychotherapy, you may rather declare that it can have positive effects on patients’ mental health.

·      What do you mean with “We recruited group of 412 subjects”? Please carefully revise your English.

·      Not all the questionnaires mentioned specifically assess body image (e.g., RSES measures self-esteem). Please, consider rephrasing.

·      As you collected additional information, please report whether you administered ad hoc questionnaires concerning socio-demographic information and BMI.

Introduction

·      Again, I would not say that aesthetic dermatology treatment is a sort of psychotherapy, you may rather declare that it can have positive effects on patients’ mental health.

·      I would like to read more data about how many people undergo aesthetic treatments and the specific situation in Poland.

·      I think it would be interesting to report the findings of studies like yours. The questions you report on page 1 lines 30-37 are very interesting and I would like to read more about literature findings.

Materials and Methods

·      You selected many useful questionnaires; I would like to read more about the process. How did you choose which constructs were worth investigating?

·      Please consider a different formula for introducing the questionnaires (i.e., “The questionnaires used” on page 3 line 91 seems confusing).

Results

·      On page 6 line 203 and page 7 line 224, rather than reporting “detailed data not shown”, please consider uploading supplementary materials with this information.

·      In general, it is not clear how you reported the scores, for instance “42.73 16.07 points”: why are there two different numbers?

·      On page 8 line 286, you mention “dependency”, but you may not infer it from correlations.

Discussion

·      On page 9 line 311, once again you mention the word “dependency”, but you did not test it.

·      For the same reason, I would use more caution when stating “We proved BMI as risk factor” (line 313). You did not prove anything; you observed a relation among variables.

·      On page 9 line 311, what do you mean with “the only conclusion which could be driven from that is rather terrifying.”? Please, provide an explanation for your findings, you subsequently formulate a question, but it remains unanswered.

Conclusion

·      I would like to read some recommendations for physicians who treat these patients. Would they benefit from a multidisciplinary approach? You may find some insights in this paper: Perego, G. & Di Mattei, V.E. (2020). A New Framework for Narcissism in Health Psychology and Psycho-Oncology. Frontiers in Psychology, 11, 1182. doi: 10.3389/fpsyg.2020.01182

·      Can you expand the sentence “The psychometric assessment of each patient before cosmetic treatment could play a significant role in choosing the appropriate approach.”? I think you are raising an interesting point and I would really like to read more about it.

General comments

·      It would be useful to have a native English speaker revise the manuscript.

Author Response

Dear Reviewer,

Thank you very much for all of your opinions and suggestions. You may find our answers attached below.

Sincerely,

Authors

Reviewer 2 Report

Manuscript jcm-1994100, titled “To inject or to reject? The body image perception among aesthetic dermatology patients” was submitted to Journal of Clinical Medicine .

This paper is an empirical examination of the rates of body dysmorphic disorder (BDD), body/shape concerns, body appreciation, and self-esteem in a sample of dermatology patients. The authors further looked at group differences in outcome variables (e.g., men vs. women, BDD vs. no-BDD). The authors had a relatively large sample and used validated measurement instruments.

I have a two primary concerns with the study as currently written. First, there was not a strong rationale for this study. Much of what was examined has been looked at in the past, and the past work was not well-summarized in the introduction. It would be beneficial for the authors to provide a more comprehensive introduction to what has already been shown, and then situate their work in the past literature, explaining what the new study will contribute, including a more specific research question. Second, the authors make comparisons between a variety of subgroups, including men versus women, people who likely have BDD vs. those who don’t, those with high vs. low self-esteem. These could be interesting comparisons, however there are significant discrepancies in the group sizes that I feel make the comparison uninterpretable. For instance, there were only 15 men compared to 397 women. It might be worth the authors’ reconsidering including those comparisons, and perhaps instead look at the sample as whole with regard to their outcome variables.

Below are some other specific comments/suggestions:

Introduction:

-the authors mention that BDD is more common in dermatology patients; it would be useful to provide the % of patients that present with BDD so that can be more directly compared to the % in the general population.

-On the whole, the Introduction is brief and lacking. Not much information is given about the past studies on BDD in dermatology patients and therefore, there is not a strong rationale for why another study is needed, beyond it being a larger sample. It would strengthen the paper significantly to delve into the past work, highlighting what is known and what could be gained from this additional study.

Method:

-the authors used a measure of body shape concerns typical of eating disorders as well as a measure of self-esteem. No information is given prior to explain why these measures were used in the study. As noted above, a more comprehensive introduction would help the reader to understand the included variables.

-similarly, it seems the authors’ main goal was to compare women and men and to compare BDD vs no BDD groups, etc. No context is given as to why these comparisons are made.

-it would be useful for the authors to include some information about the reliability of their measures in the current study (e.g., coefficient alpha) as well as more information about the translations of the measures. For instance did the Polish translations of the measures already exist or was the translation done for this study in particular? If the latter, what was the process for the translation?

Results:

-I have significant concerns about the discrepancy in group sizes – comparing a group of 397 women to 15 men does not seem reasonable and is likely subject to significant error. I would be hesitant to interpret any such comparison results.

-I have similar concerns about the sample size discrepancy between the BDD (30) and non-BDD (382) groups and the size/shape concern groups.

Discussion:

-it seems that most of the findings are similar to past work. Replication is good; I also encourage the authors to talk more about the practical applications of their work. For instance, what do these results suggest dermatology practices should screen for? What are the implications for practices if they do have clients that screen high for these risk factors? Should those clients be denied surgery/procedures until they receive counseling, for instance?

-a specific example of where more could be said is the finding that income/financial status was a robust predictor of negative outcome. What are the implications of this? For instance, do we need to focus more on prevention in low-income areas? There is other literature that finds similar results for other types of disorders as well, which could be brought into the discussion.

Other:

-the paper would benefit from further proofreading

Author Response

(The authors gave the same response as above.)

Round 2

Reviewer 2 Report

Thank you for the opportunity to review this revised manuscript. I appreciated the authors’ response to the reviewers’ comments and the additions they made to the manuscript. I have only a few remaining comments.

 -The authors have strengthened the introduction by stating what is novel about their own study. It would still be useful to provide more specific information at the end of the Introduction on what the actual research question and hypotheses were. A clearer statement that they will be comparing men to women and BDD to non-BDD would help to contextualize the analyses and link to past work.

 -It is helpful that the authors have now included the statement about the measures being available in Polish. I still think it would be useful to include specific reliability information on each measure – for instance, what is the coefficient alpha of each measure in the current study?

 -In the authors’ reply, they said that their goal was not to compare men and women and BDD vs. non-BDD, however the first two sets of results presented do exactly that. Indeed, in the paper, they state “For the need of the research interpretation the studied individuals were divided into the BDD and the non-BDD group”. So I am not sure I understand the reply. As noted above, this aspect of the analyses could be referenced in the Introduction.

 -I appreciate that the authors noted that the male group (n=15) may be too small to draw inferences from. However, they state that the BDD group (n=30) is of sufficient size to compare to the non-BDD group (n=382). This still seems like a very big discrepancy in group sizes for statistical comparison and it seems worth mention in the paper (or cite sources to show that this value is appropriate).

Author Response

Dear Rewiever,

thank you very much for all your suggestions.

We corrected our manuscript accordingly.

Please find the attached file below.

Sincerelly,

Authors
